# Intelligent Fault Diagnosis of Liquid Rocket Engine via Interpretable LSTM with Multisensory Data

**DOI:** 10.3390/s23125636

**Published:** 2023-06-16

**Authors:** Xiaoguang Zhang, Xuanhao Hua, Junjie Zhu, Meng Ma

**Affiliations:** 1Xi’an Aerospace Propulsion Institute, Xi’an 710100, China; zhangxiaoguang0522@163.com; 2School of Future Technology, Xi’an Jiaotong University, Xi’an 710049, China; 3School of Mechanical Engineering, Xi’an Jiaotong University, Xi’an 710049, China; zjj_appreciate@stu.xjtu.edu.cn (J.Z.); meng_ma@xjtu.edu.cn (M.M.)

**Keywords:** interpretable, bidirectional LSTM, data fusion, fault simulation

## Abstract

Fault diagnosis is essential for high energy systems such as liquid rocket engines (LREs) due to harsh thermal and mechanical working environment. In this study, a novel method based on one-dimension Convolutional Neural Network (1D-CNN) and interpretable bidirectional Long Short-term Memory (LSTM) is proposed for intelligent fault diagnosis of LREs. 1D-CNN is responsible for extracting sequential signals collected from multi sensors. Then the interpretable LSTM is developed to model the extracted features, which contributes to modeling the temporal information. The proposed method was executed for fault diagnosis using the simulated measurement data of the LRE mathematical model. The results demonstrate the proposed algorithm outperforms other methods in terms of accuracy of fault diagnosis. Through experimental verification, the method proposed in this paper was compared with CNN, 1DCNN-SVM and CNN-LSTM in terms of LRE startup transient fault recognition performance. The model proposed in this paper had the highest fault recognition accuracy (97.39%).

## 1. Introduction

Liquid rocket engines (LREs) are the highest energy systems as they produce up to thousands kN of thrust. The energy levels reach up to several GW of power by converting the high energy of the combustion product into high speed ejected mass flows [1]. To provide the high energy levels, LREs have to operate in a harsh thermal and mechanical environment, which may result in catastrophic anomalies due to unexpected events. Therefore, the fault diagnosis system has been a fundamental key to safety and reliability of LREs. In addition, for reusable LREs, fault diagnosis is able to provide the condition of systems. If a fault has been detected, the engine will be closed in case of catastrophic failures in launch vehicle systems. A lot of fault diagnosis algorithms and frameworks for space shuttle main engine (SSME) has been developed. For example, red-line cutoff system [2,3], Health Monitoring system (HMS), Advanced Health management system (AHMS) [4,5]. These methods have greatly improved the reliability of LREs.

In general, the methods for fault diagnosis can be categorized into data-driven approaches and model-based approaches. Today, data-driven models are advancing by leaps and bounds with present computing power and algorithms. These technologies include not only regression analysis, EM (Expectation-Maximum) algorithm, Bayesian theory [6,7], and other classical probability and statistical methods, but also SVM [8,9] and other classical machine learning methods. Liu et al. established an adaptive correlation algorithm and envelope method for real-time fault detection and alarm during steady-state and startup processes of LRE [10]. Similarly, deep learning and artificial intelligence technologies are increasingly applied to the fault diagnosis of liquid rocket engines [11,12,13]. Flora, J.J. et al. developed an artificial neural network-based isolation and replacement algorithm for the fault management of LRE sensors [11]. Wen et al. used a conversion method that converts signals into two-dimensional (2-D) images to extract the features of the converted 2-D images and eliminate the influence of handcrafted features [14]. Chen et al. proposed a physics-informed deep neural network based on multi-sensor signals for bearing prognosis in liquid rocket engine fault diagnosis, providing a new approach [15]. Similarly, Wang et al. published a study on intelligent fault diagnosis of planetary gearboxes using transferable deep Q networks in the same journal, providing a new technical solution for liquid rocket engine fault diagnosis [16]. These studies provide new ideas and methods for the application of deep learning and artificial intelligence technologies in the field of rocket engine fault diagnosis, offering more choices for liquid rocket engine fault diagnosis. Though these methods utilize the powerful learning capabilities of deep learning models, they ignore the interpretable information. The model-based approaches utilize LRE’s model to determine the parameters through Kalman Filters. Lee built a mathematical model of an open-cycle liquid propellant rocket engine and artificially injected different kinds of faults, then Kalman filter and fault factor methods were used for fault diagnosis [17]. The performance of model-based methods relies on the accuracy of mathematical models. However, developing a precise mathematical model is challenging because of the complicated structure. The data-driven approaches take advantage of monitoring data to detect the fault through machine learning methods.

In this study, we used numerical models to construct data sets containing potential failure types during engine start-up, and planned to use deep neural networks for targeted training. At the same time, we prefer to choose a machine learning model with high interpretability rather than a black box model with high decision risk, so a novel method based on 1D-CNN and interpretable bidirectional LSTM (1D-CNN-iBLSTM) for fault diagnosis of LREs is proposed. 1D-CNN is used for multi-variable features extraction, then an interpretable bidirectional LSTM is designed to model the sequential features extracted through 1D-CNN, which improves the performance of fault diagnosis. Several LRE system simulations were carried out to generate fault and healthy data. Based on the simulated data, the proposed method is used for fault diagnosis. The contribution of this study is summarized as follows:

(1) A novel 1D-CNN and interpretable LSTM is proposed for LREs’ fault diagnosis, where 1D-CNN is responsible for features extraction with multi-variables. An interpretable LSTM is constructed for modeling the sequential features.

(2) The simulated datasets containing normal and fault states are generated through system simulation of LREs. 1D-CNN and interpretable LSTM is used for fault diagnosis. The results demonstrate that the proposed method produced very low false alarm rates and low missed detection rates.

The remainder of this study is organized as follows: Section 2 introduces system simulation of LRE, where simulated data is generated. Section 3 describes the main concepts used in this paper and presents the proposed method. The fault diagnosis and discussion are shown in Section 4. In Section 5, we summarized the results and draw the conclusion.

## 2. Simulation System Construction

### 2.1. System Simulation of LRE

This section introduces the construction of a simulation system for a liquid rocket engine, namely SSME. Firstly, the idea of hierarchical structure modeling based on structural composition and working process is established. Both normal state and fault state are simulated to generate the datasets, where fault mode library is determined according to [18]. Secondly, in order to achieve modular fault simulation of the engine system, a corresponding software system was designed. Finally, various fault simulations were performed on the rocket engine [19,20]. The engine structure diagram is shown in Figure 1.

### 2.2. Fault Simulation of LRE System

Based on the simulation model, various possible fault modes of a large-thrust hydrogen-oxygen engine were dynamically and comprehensively analyzed by injecting faults into the system through methods such as adding or reducing modules, modifying modules, and key performance parameters in the modules [17,21]. Table 1 lists the major fault modes and their corresponding manifestations of engine components. We selected valve opening fault, hydrogen turbine flow leakage, cooling jacket leakage, and turbine component efficiency decrease as representative faults for simulation studies. Further details regarding the construction of these faults are presented as follows.

#### 2.2.1. Valve Opening Failure

Valve control is a critical factor in the normal start-up of an engine. By adjusting the timing and response speed of five main valves, namely the main oxidizer valve (MOV), main fuel valve (MFV), fuel pre-burner oxidizer valve (FPOV), chamber coolant valve (CCV), and oxidizer pre-burner oxidizer valve (OPOV), valve failures can be simulated [22]. The specific formula is as follows, which mainly adjusts the flow rate through the control function to simulate valve failures such as valve not opening, slow valve opening, and valve blockage.
(1)$$\dot{m}=c_{q}A\tau \sqrt{2\rho \Delta p}$$
where $\dot{m}$ is the flow rate through the valve, $c_{q}$ is the flow coefficient, A is the maximum flow area, τ is the control function, ρ is the average density of fluid flowing through the valve, and Δp is the pressure difference between the two ports of the valve.

#### 2.2.2. Hydrogen Turbine Leakage

Hydrogen, as a fuel, is relatively easy to leak due to its small molecular weight. In addition, the rotational speed of the hydrogen turbo pump can reach tens of thousands of revolutions per minute, and the turbo pump is a coaxial structure, with the pressure at the turbine end higher than that at the pump end, making it easy for hydrogen to leak into the pump and other environments [23]. In this fault mode, for the engine system, liquid hydrogen leaks directly into the pump and environment, which is equivalent to adding two flow paths. A valve component with a maximum flow area of A is added to each flow path, and the valve opening size is controlled by an external signal to characterize the severity of the leak. The specific formula is as follows:(2)$${\dot{m}}_{3}={\dot{m}}_{1}+{\dot{m}}_{2}$$
(3)$${\dot{m}}_{2}=c_{q}A\tau \sqrt{2\rho \Delta p}$$
where ${\dot{m}}_{3}$ and ${\dot{m}}_{1}$ represent the main flow, ${\dot{m}}_{2}$ is the leakage flow in the pipeline, and A is the maximum flow area of the leakage pipeline.

#### 2.2.3. Cooling Jacket Leakage

Similar to hydrogen turbine leakage, after passing through the high-pressure hydrogen turbine pump and cooling jacket, liquid hydrogen forms high-temperature and high-pressure hydrogen gas, which can easily leak into the thrust chamber and participate in combustion. For the engine system, hydrogen leakage from the cooling jacket to the combustion chamber is equivalent to adding a new flow path. By adding corresponding valve components and setting the size of the valve opening, the severity of the leakage can be characterized. The valve opening is controlled by an external signal.

#### 2.2.4. Turbine Component Efficiency Decrease

During operation, turbine components may experience faults such as rotor rubbing or sticking, shaft fracture, turbine blade detachment, pump blade fracture, and oxygen pump cavitation, which all result in a decrease in turbine component efficiency. Therefore, an efficiency correction factor is introduced to change the efficiency of the turbine and centrifugal pump, simulating a decrease in power that leads to a decrease in speed, a reduction in the work done by the centrifugal pump, and thus achieving fault simulation [24,25]. The specific formula is as follows:(4)$$P_{turbine}=dpQ\eta_{turbine}f=n_{turbine}T$$
where $P_{turbine}$ represents power, dp is the pressure difference across the turbine, Q is the volume flow rate, η is the efficiency of the turbine, f is the correction factor, $n_{turbine}$ is the common rotational speed of the turbine and centrifugal pump, and T represents the torque.

## 3. Methodology

### 3.1. Recurrent Neural Network (RNN) and Long Short-Term Memory (LSTM)

Figure 2 shows the model of recurrent neural networks. An important advantage of recurrent neural networks (RNN) is the ability to use context-dependent information in the mapping between input and output sequences.

Unfortunately, standard recurrent neural networks (RNN) have a limited range of contextual information to access. This problem makes the influence of hidden layer input on network output decline with the continuous recursion of network loop. Thus, to solve this problem, the LSTM structure was born. Rather than being a type of recurrent neural network, LSTM is a reinforced version of the component placed within the recurrent neural network. To be specific, it is to replace the small circle in the hidden layer of the cyclic neural network with the module of short-term memory development. As shown in Figure 3, main structure of LSTM network includes:
Forget gate: The forgetting gate decides what information to discard. The input is the calculation result of the previous neuron *S_t−_*_1_ and the current input vector *x_t_*. After the two are joined and passed through the forgetting gate ($Sigmoid\left(x\right)=\frac{1}{1+e^{-x}}$ will decide what information to keep and what information to discard), a 0–1 vector (the dimension is the same as the output vector *C_t−_*_1_ of the previous neuron) is generated (See Equation (5)). When the vector is dotted with *C_t−_*_1_, the information retained by the previous neuron after calculation is obtained, which determines how much *C_t−_*_1_ is kept in *C_t_*.
(5)$$f_{1}=sigmoid\left(\omega_{1}\left[\begin{array}{c} S_{t-1}\\ x_{t} \end{array}\right]+b_{1}\right)$$Input gate: Represents information to be saved or information to be updated. As shown in the Figure 3b, it is the connection vector between *S_t−_*_1_ and *x_t_*. The result obtained after passing through the sigmoid function is the output result of the input gate, which determines how much information from *x_t_* can be used to calculate cell state *C_t_*.
(6)$$f_{2}=sigmoid\left(\omega_{2}\left[\begin{array}{c} S_{t-1}\\ x_{t} \end{array}\right]+b_{2}\right)\times \tanh\left({\hat{\omega }}_{2}\left[\begin{array}{c} S_{t-1}\\ x_{t} \end{array}\right]+{\hat{b}}_{2}\right)$$The update status of a new cell is shown in Equation (7).
(7)$$C_{t}=f_{1}\times C_{t-1}+f_{2}$$Output gate: The output gate determines the hidden vector *S_t_* of the current neurogenic cell output. Different from *C_t_*, *S_t_* is a little more complicated. It is the multiplication product of the computed tanh(Ct) with the computed result of the input gate, which is described by the formula as shown in Equation (8).
(8)$$S_{t}=sigmoid\left(S_{t-1}\right)\cdot \tanh\left(C_{t}\right)$$


### 3.2. Bidirectional LSTM

Bidirectional LSTM is another variant of recurrent neural networks (RNNs) that considers both past and future information at each time step. In the structure diagram shown in Figure 4, we detail the structure of the bidirectional LSTM module. In traditional unidirectional LSTM, only past information before the current time step is considered, while future information is ignored. This unidirectional model may be limited by long-term dependencies and struggle to handle long sequences [26].

To address this issue, the Bidirectional LSTM model introduces another LSTM network that reads the input sequence in the opposite direction at each time step. This enables the model to consider both forward and backward information simultaneously, leading to better handling of long sequences [27].

Specifically, the computation process of Bidirectional LSTM is as follows:The forward LSTM reads the input sequence in chronological order and computes the hidden state vector for each time step.The backward LSTM reads the input sequence in the reverse order and computes the hidden state vector for each time step.The hidden state vectors of the forward and backward LSTMs are added element-wise to obtain the final hidden state vector for each time step.

In practical applications, Bidirectional LSTM is often used in natural language processing (NLP) tasks, such as sentiment analysis, language modeling, fault diagnosis, etc.

### 3.3. Interpertable LSTM Based on Attention Mechanism

Attention mechanism is a commonly used mechanism in deep learning that is mainly used to weight the importance of different parts of the input data so that the network can better focus on important parts [28]. Figure 5 shows the most common attention frame. The Attention mechanism is based on sequence models such as Recurrent Neural Networks (RNNs) and Long Short-Term Memory Networks (LSTMs), and is often used in conjunction with Convolutional Neural Networks (CNNs) [29,30,31].

The principle of the Attention mechanism is to encode the input data to generate a set of feature vectors, and then determine the importance of each feature vector by calculating the similarity between each feature vector and a specific “attention weight” vector. These weights can be viewed as coefficients used to calculate weighted sums [32]. In this way, the Attention mechanism can make the network pay more attention to important features, thereby improving the performance of the model. Generally speaking, the Attention mechanism can be divided into the following steps:The encoder encodes the input data to generate a set of feature vectors.Calculate the similarity between each feature vector and a specific “attention weight” vector to determine the importance of each feature vector.Multiply the attention weights with the feature vectors and add the results to obtain a weighted feature vector representation.Use the weighted feature vector as input to the next layer and repeat the above steps.Finally, add all the weighted feature vectors to obtain a comprehensive representation for the final prediction.

Overall, the Attention mechanism allows the network to adaptively select important information in the input sequence, thereby improving the performance of the model. It has been successfully applied to various deep learning tasks such as natural language processing, image processing, and time series prediction.

### 3.4. Spatial Attention Operation

Assuming that there is a 2D spatio-temporal feature matrix $X\in R^{N_{s}\times N_{T}}$, in which $N_{s}$ represents the number of features (number of sensor signals) in a single time step and $N_{T}$ is the number of time steps, so the input feature matrix X can be divided into $N_{T}$ $N_{s}$-dimension vectors ($x_{t}\in R^{N_{s}\times 1}$). The calculation process of spatial attention weight is shown in Figure 6. After passing through a fully connected layer (Dense), a sigmoid function is used to activate the input feature vector. Then, the $Softmax\left(x_{i}\right)=\frac{e^{-x_{i}}}{\sum_{i=1}^{n}e^{-x_{i}}}$ function normalizes each element in several vectors of input so that the sum is 1 (See Equation (9)). These vectors all contain some spatial attention weights, which are used to calculate a weighted average to determine which elements in the input sensor sequence should receive more attention, that is, to dynamically focus on important spatial features [33,34]. The final output vector is the Hadamard product of $\alpha_{t}$ and $x_{t}$ like Equation (10).
(9)$$\alpha_{t}=f_{S-A}\left(x_{t}\right)={\left[\alpha_{1}^{t},\alpha_{2}^{t},\ldots ,\alpha_{N_{s}}^{t}\right]}_{N_{s}\times 1}$$
(10)$$x_{t}^{\prime }=\alpha_{t}⊙\;x_{t}={\left[\alpha_{1}^{t}x_{1}^{t},\alpha_{2}^{t}x_{2}^{t},\ldots ,\alpha_{N_{s}}^{t}x_{N_{s}}^{t}\right]}_{N_{s}\times 1}$$

### 3.5. Temporal Attention Operation

Similarly, as shown in Figure 7, we use the same approach to apply attention module 2 after the LSTM layer to focus on more important temporal information. Suppose the hidden layer state output sequence is obtained successively as Equation (11) shown, then inputted into the T-A model after being transposed to get the temporal attention weight β (See Equation (12)). Afterward, the final output vector is the matrix product of $\beta^{T}$ and $H^{T}$ like Equation (13).
(11)$$H={\left[h_{1},h_{2},\ldots ,h_{N_{T}}\right]}_{s\times N_{T}}$$
(12)$$\beta =f_{T-A}\left(H\right)={\left[\beta_{1},\beta_{2},\ldots ,\beta_{N_{T}}\right]}_{N_{T}\times 1}$$
(13)$$h_{att}=\beta^{T}⊗H^{T}=\overset{N_{T}}{\underset{i=1}{\sum }}\beta_{i}h_{i}\;,\;h_{att}\in R^{1\times s}$$

### 3.6. The Proposed Fault Detection Framework

Figure 8 shows the comparison of partial data between normal mode and failure mode of LRE. This figure shows the change values of 28 state monitoring parameters in three stages of LRE: starting, steady state (including variable operating conditions) and shutdown. Figure 8a shows the comparison of single parameters in normal and fault states, and Figure 8b shows the comparison of overall parameters, curves in different colors represent different state parameters. The ultimate goal of employing data-driven fault analysis techniques is to achieve automatic fault diagnosis, reducing the reliance on experts’ experience and knowledge for offline analysis. This approach involves using measured data over time to classify the system’s state and identify the root cause of failure, bridging the gap between traditional diagnostic methods and automated fault detection and diagnosis [35].

One promising approach for automatic fault diagnosis is the utilization of deep learning techniques, such as Convolutional Neural Networks (CNNs) and Bidirectional Long Short-Term Memory (Bidirectional LSTM) networks [32,36]. The combination of CNN and LSTM allows the model to extract local features from input sequences through convolutional operations and capture long-range sequence dependencies through LSTM memory cells, enhancing the model’s feature extraction capability. Additionally, the parallel computation of CNN accelerates the training process, addressing the potential issue of slower training speed in LSTM layers. By leveraging the advantage of parallel computation, the combination of CNN and LSTM speeds up the training process of the model. Moreover, LSTM, as a type of gated recurrent unit, effectively alleviates the problem of gradient vanishing in deep neural networks [37,38,39]. The combination of CNN and LSTM introduces more gradient pathways between different layers of the model, helping to mitigate the issue of gradient vanishing and improve the training stability of the model. However, it is important to note that the key to LSTM’s ability to capture long-term dependencies is to store each step’s input information in the memory cell. Each output hidden state contains all the input information up to the current time step. As hidden states are typically represented by vectors of fixed length, the network gradually compresses all the information over time. However, this indiscriminate compression can weaken the time differences between input features to some extent and may fail to highlight crucial information in the history. Therefore, appropriate improvements are necessary to enhance the discriminative power of LSTM.

These neural network architectures are capable of processing sequential data and capturing both spatial and temporal features from the measured data. For example, in the context of fault diagnosis, a CNN-LSTM model can be designed to take in time-series data as input, where the CNN component extracts spatial features from the data, and the LSTM component captures temporal dependencies. The data is input into the LSTM in units of time steps to generate a series of hidden states. Then, self-Attention is used to weigh and sum the hidden states of historical fault information to obtain a context vector, which represents the correspondence between current sensor data and historical fault information. Finally, this context vector is combined with current sensor data and operational status information and input into the output layer to generate results. The advantage of this method is that it can automatically learn the correspondence between historical fault information and current sensor data and model the correspondence between different time steps, thereby improving the accuracy and reliability of fault diagnosis. This CNN-LSTM model can be trained on labeled data, including examples of normal operating conditions and different fault scenarios. By learning from this labeled data, the model can automatically identify patterns and correlations in the data indicative of specific fault types.

This data-driven approach allows for the development of a fault diagnosis system that can automatically detect and classify faults in real-time, without the need for expert intervention. This can significantly reduce the time and effort required for fault diagnosis, leading to improved system reliability and reduced downtime in various applications, such as industrial manufacturing, power systems, and transportation.

Compared to 2D-CNN, 1D-CNN typically has lower model complexity, as it only needs to consider feature extraction from one-dimensional data without considering the height and width of two-dimensional image data. This makes it more suitable for scenarios with limited computational resources, such as devices with small memory or embedded systems. Moreover, the sensor parameters of mechanical systems are typically one-dimensional sequential data, such as temperature and pressure, with data points collected over time forming a one-dimensional vector. Therefore, using 1D-CNN can naturally handle such one-dimensional sequential data without introducing the two-dimensional image structure, reducing the complexity and memory footprint of the model. Additionally, for certain fault detection tasks in mechanical systems, where the number of fault samples may be limited and not sufficient to support the training of 2D-CNN with a large number of parameters, 1D-CNN as a simpler model structure can still achieve good performance even in small sample situations [40].

## 4. Fault Diagnosis

### 4.1. Overall Model Analysis of Fault Diagnosis

To diagnose the startup transient fault in LRE, we adopt a 1D-CNN-ALSTM architecture due to the large volume of sample data and limited sample quantity, as illustrated in Figure 9. The attention weight calculation process of attention module 1 is shown in Figure 6 and Figure 7. First, we stack N_s_ (=28) time series of each sensor data between 0 and tf (=2 s), constructing a 2D array (N_T_ × N_s_) containing the test data, where N_T_ (=tf/dts) is the number of sample points during the launch period [41], as illustrated in Figure 10. Due to the different dimensions and orders of magnitude of data, we standardized the data with zero-mean normalization, keeping all dimensions the same weight (because each dimension follows the normal distribution with the mean value of 0 and variance of 1). In the final calculation of distance, the data of each dimension plays the same role. The selection of different dimensions can avoid the great influence on distance calculation.

The original time-series data is too complex to be directly input into the neural network for processing. Therefore, we plan to divide the original time series data into multiple subsequences, each with the same and shorter length, which is convenient for the neural network to process and learn, namely sliding window operation. At the same time, each subsequence contains a part of the original time series data, allowing for feature extraction for each subsequence and extraction of the important features of each subsequence. These features are then used as input to the neural network to further improve the accuracy and efficiency of the model.

The sliding window operation is particularly useful when dealing with time series data, as it allows for long sequence feature data to be transformed into multiple short sequence feature data, which can be better processed. Although LSTM can handle long sequence data, if the sequence is too long, it can increase the difficulty of LSTM training. By using the sliding window operation to divide the data into shorter subsequences, we can mitigate this problem and ensure that the data can be processed more efficiently. In this case, the size of the array is N_Tk_ × N_s_, and the number of windows to be prepared is N_f_ = (N_T_ − N_Tk_)/N_d_ + 1 [42]. By adjusting the size of the sliding window and the stride length, we can control the number of subsequences generated, allowing us to balance the model’s accuracy and complexity.

We prepared N_f_ CNNS to extract the features of each slice window separately, get N_f_ feature sequences at different time points, and then splice these sequences together. BiLSTM layer learns the temporal dependencies provided by multiple feature maps extracted in parallel by CNNs. The Softmax layer after the fully connected layer retrieves the probability distribution of failure modes.

We use rectified linear unit (ReLU) as the activation function for CNN-LSTM to prevent gradient vanishing. Additionally, we apply max-pooling layers to reduce the size of the output data (activation maps) and emphasize specific data received from the convolutional layers. After evaluating different combinations and considering the impact on classification performance and computation time, we choose 2 layers of CNN and 1 layer of Bidirectional LSTM. To train our model, we use cross-entropy loss as the cost function for multi-class classification and apply adaptive moment estimation for stochastic gradient descent optimization of weights and biases in the training dataset [43,44]. We implement the training and testing using GPU acceleration in TensorFlow and Keras, which allows us to accelerate the training process and improve the efficiency of our model.

### 4.2. Result Analysis

After training our model, we evaluate its performance on a test dataset, which consists of 20% of the complete dataset that was not used for training. Figure 11 and Figure 12 show the change curves of the loss value of the model training and verification sets and the accuracy rate change curves of the training and verification sets.

It can be seen from the change curves of the loss value of the training set and the verification set in Figure 10 that with the increase of epoch, the loss value of the training sample and the verification sample keeps decreasing and eventually tends to be relatively stable. In the first 16 iterations, the loss value of the training sample and the validation sample decreased very fast, and the decline rate of the two basic curves continued to change synchronously. The loss value decreased from around 2.3 to around 0.4, indicating that the model is rapidly converging. Within 16 to 32 iterations, the rate of descent of the training and validation samples slowed down significantly compared to the first 16 iterations. The loss value dropped from about 0.4 to about 0.25, indicating that the model is still learning and has a tendency to converge. After 32 iterations, the loss values of the training samples and verification samples gradually approached 0. The two curves also basically overlapped, and the loss value does not change further. This indicated that the model had completed training and had good convergence.

As can be seen from the change curves of the accuracy of training set and verification set shown in Figure 11, within the first 16 iterations, the accuracy of training samples and verification samples increased rapidly and fluctuated significantly, from about 25% to about 90%. Between 16 and 32 iterations, the accuracy of training samples and validation samples increased relatively steadily and slowly, from about 90% to 100% accuracy. After 32 iterations, the accuracy of training samples and verification samples did not change again and remained at 100%. This indicated that the network model had been trained and also proved that this model had fast convergence and high accuracy of fault diagnosis and classification.

The results of the classification performed by the CNN-LSTM model are presented in Figure 13. The confusion matrix shows the classification results for 4 classes (3 failure modes and 1 normal state) at tf = 2 s, which indicates perfect accuracy. The classification accuracy for the three failure modes and one normal state is good, which suggests that our model is effective for fault diagnosis in LRE startup transients. Setting the range of the sliding window N_Tk_ from 0.8 s to 1.4 s, after multiple tests, it was found that the CNN-LSTM network performs best in terms of fault classification response time when N_Tk_ is set to 1.2 s. In addition, the shorter the duration of the stride window N_d_, the higher the prediction accuracy, but the training cost increases with the number of windows (N_f_), so we choose N_d_ as 0.2 s.

Considering that the number of sample points is too large, and the data of startup phase and shutdown phase have little influence on the results of fault analysis and diagnosis, we extract data within a 1 s interval containing the moment of fault occurrence as new samples, and retrain the model with a sliding window N_Tk_ of 0.6 s and a stride window duration N_d_ of 0.1 s, greatly reducing the amount of data used for training and significantly reducing the training time while maintaining the classification accuracy.

### 4.3. Comparative Analysis

Finally, in order to further verify the superiority of the 1DCNN-A-BiLSTM model, we used the CNN, 1DCNN-SVM and CNN-LSTM models for comparison and verification. The average value was obtained for each run 10 times, and the final diagnostic results and standard deviations are shown in Table 2.

Under the same experimental conditions, four fault diagnosis methods were compared. The CNN model puts the original signal directly into a two-dimensional convolutional neural network to identify and classify the faults. The 1DCNN-SVM model puts the original signal into a one-dimensional convolutional neural network for feature extraction and then the support vector machine is used in place of the Softmax layer for fault recognition and classification. The CNN-LSTM model puts the original signal into a two-dimensional convolutional neural network for feature extraction and then and then the feature sequence is sent into the LSTM for further feature extraction. Finally, the output result is obtained by the Softmax layer. As can be seen from Table 2, by comparing the accuracy of the four fault diagnosis methods, the fault recognition accuracy of the 1DCNN-ABiLSTM model proposed in this paper reaches 97.39%. Compared with CNN, 1DCNN-SVM and CNN-LSTM, our method increases the accuracy rates by 10.40%, 3.56% and 2.63% respectively. By comparing the diagnostic results of various fault diagnosis methods, it is shown that the proposed model has the highest test accuracy, the lowest standard deviation and better time performance. The lifting effect has obvious advantages and is more suitable for application to diagnose startup transient fault in LRE.

## 5. Discussion

By using a combination of CNN and LSTM and implementing the sliding window operation, we have developed an accurate and efficient fault diagnosis system that can automatically detect and classify faults in real-time without the need for expert intervention. This can significantly reduce the time and effort required for fault diagnosis, leading to improved system reliability and reduced downtime in various applications. During the startup process of a rocket engine, different sensor data changes over time, and the Attention mechanism can dynamically focus on important features at different time points. Therefore, it can help the model better capture key features at different time points and improve the accuracy of fault diagnosis. At the same time, because it can clearly indicate which features are most important for fault diagnosis at different time points, this is very useful for engineers and technicians because it can help them better understand the working principle of liquid rocket engines, and how to optimize and improve the performance and reliability of rocket engines.

In summary, applied with the Attention mechanism, the use of a combination of CNN and LSTM along with the implementation of the sliding window operation has resulted in the development of an accurate and efficient fault diagnosis system. This system can detect and classify faults without the need for expert intervention, thereby significantly reducing the time and effort required for fault diagnosis. The improved system reliability and reduced downtime make this system useful in various applications.

## 6. Conclusions

In this paper, we propose a liquid rocket engine fault diagnosis model (CNN-LSTM) based on the attention mechanism. After verification, the attention-based CNN-LSTM model outperformed both the LSTM model and the CNN model in experiments. The CNN-LSTM model with both spatial and temporal attention modules perform best among the models we used, highlighting the benefits of the proposed spatio-temporal attention. Inspired by the performance of the GCN model, our next study will consider the use of LRE’s startup transient spatial graph information to further improve the performance of the model. Current and future works are aimed at launch data augmentation, and physical interpretation of data-driven models.

## Figures and Tables

**Figure 1 sensors-23-05636-f001:**
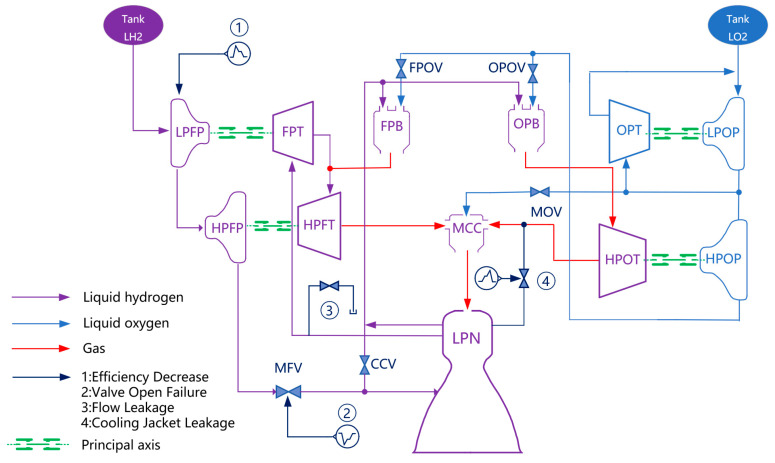
Schematic of LRE and selected failure modes.

**Figure 2 sensors-23-05636-f002:**
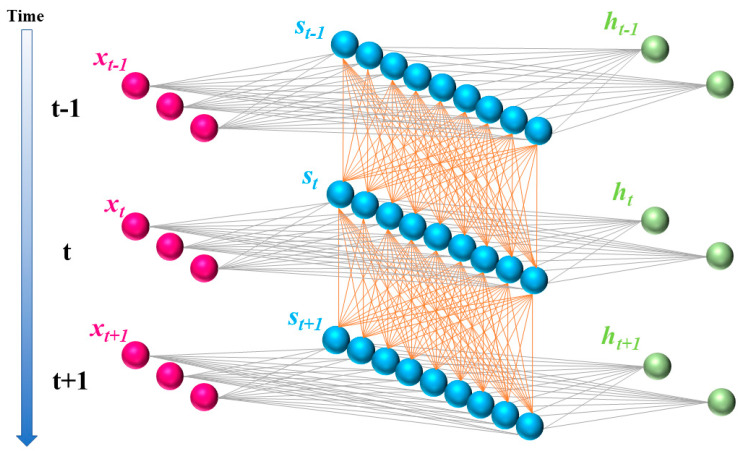
The structure of RNN.

**Figure 3 sensors-23-05636-f003:**
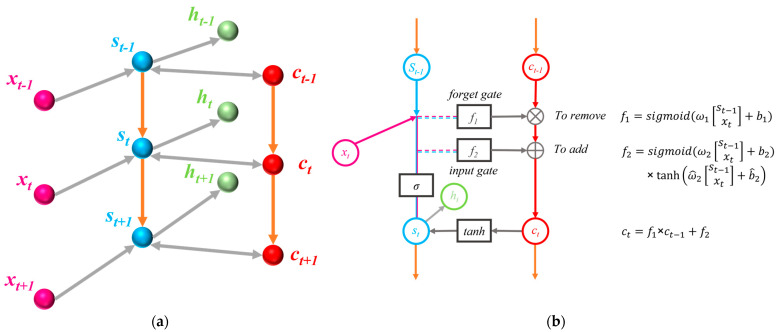
The structure of LSTM: (**a**) 3D structure of LSTM; (**b**) Schematic diagram of LSTM.

**Figure 4 sensors-23-05636-f004:**
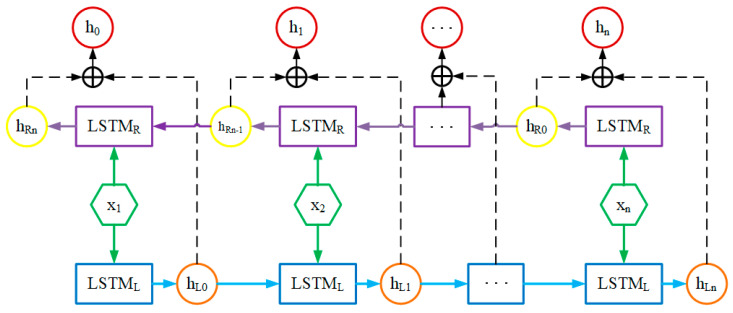
The structure of Bidirectional LSTM.

**Figure 5 sensors-23-05636-f005:**
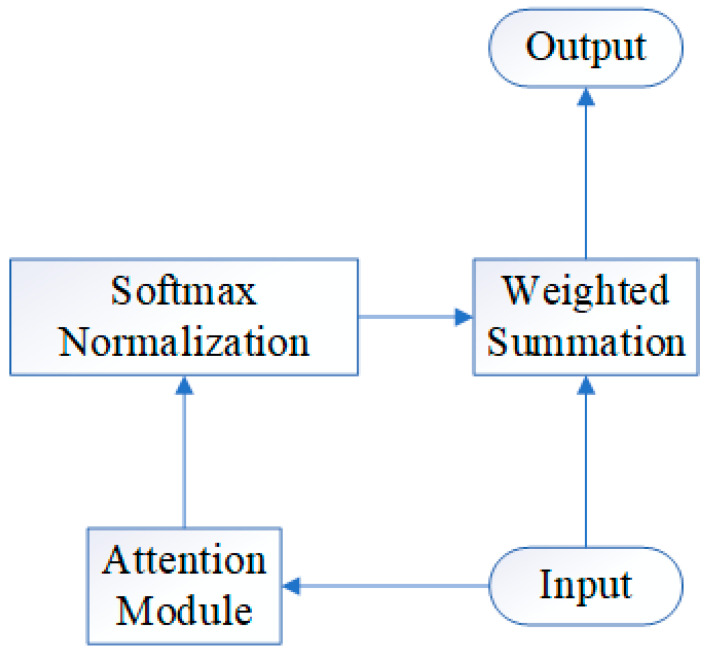
Illustration of attention mechanism.

**Figure 6 sensors-23-05636-f006:**
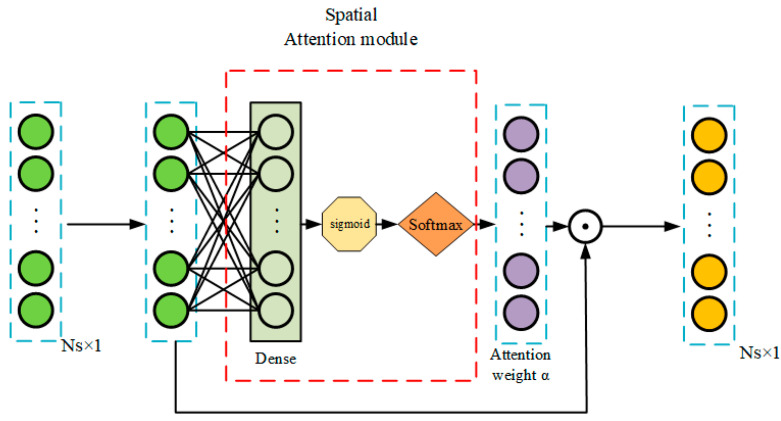
Illustration of the spatial attention operation.

**Figure 7 sensors-23-05636-f007:**
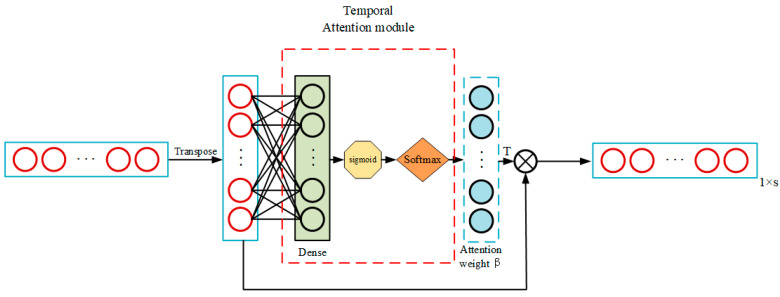
Illustration of the temporal attention operation.

**Figure 8 sensors-23-05636-f008:**
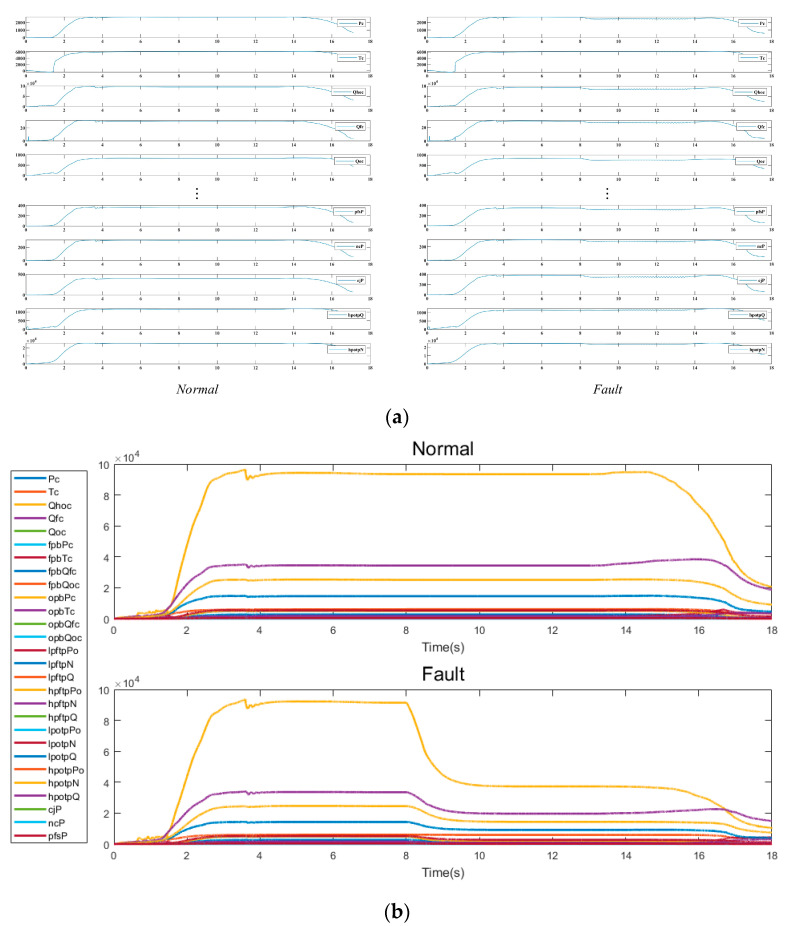
Comparison of partial data between normal mode and failure mode of LRE: (**a**) Comparison of different parameters (**b**) Single instance comparison chart.

**Figure 9 sensors-23-05636-f009:**
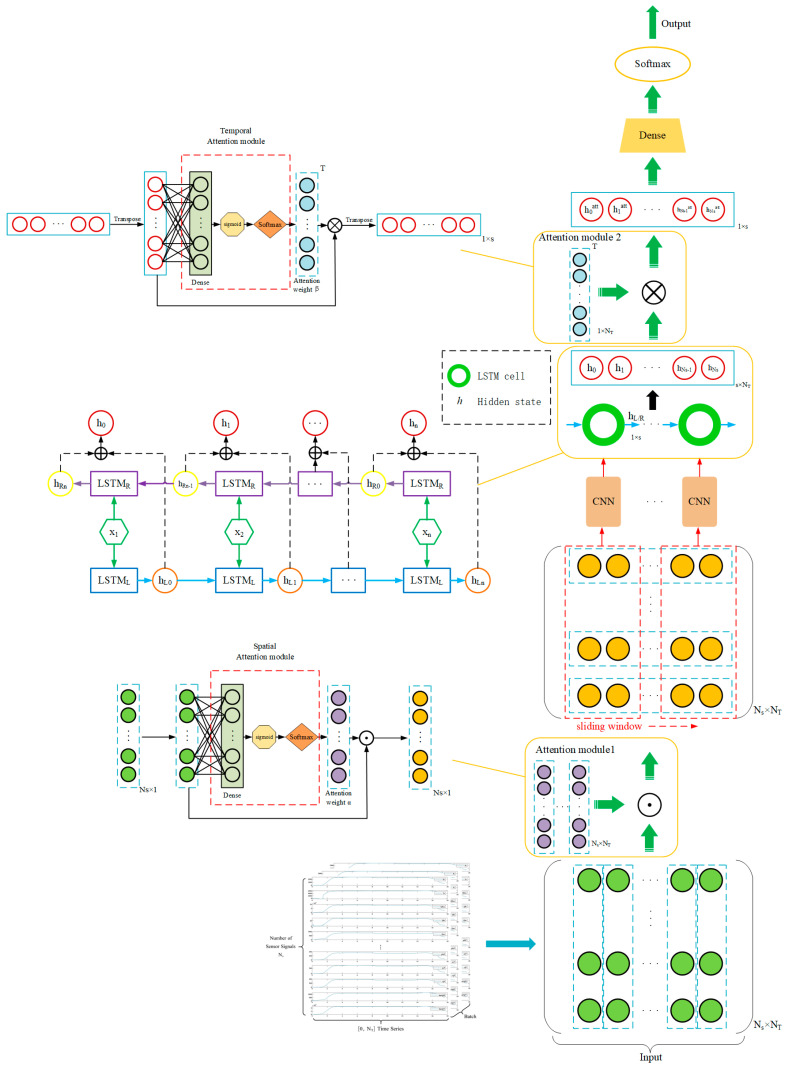
1D-CNN–ALSTM architecture for fault diagnosis.

**Figure 10 sensors-23-05636-f010:**
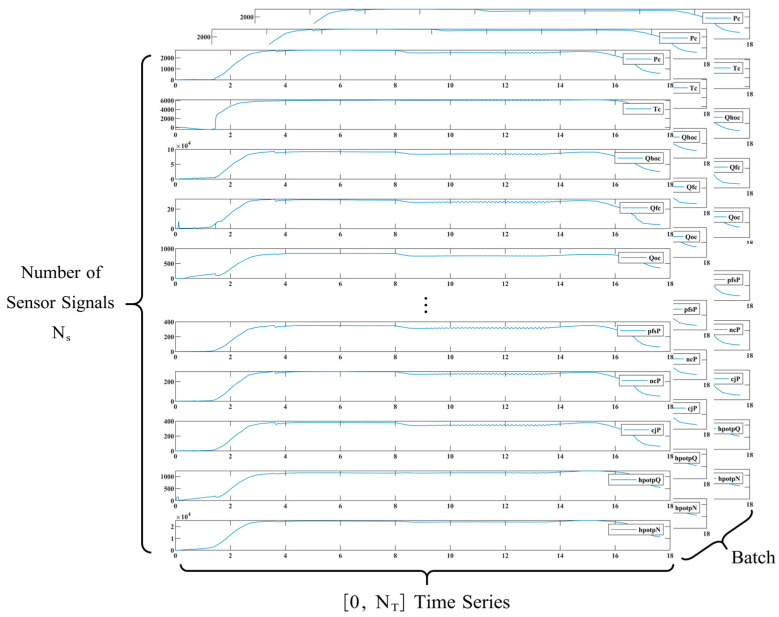
Composition of the dataset.

**Figure 11 sensors-23-05636-f011:**
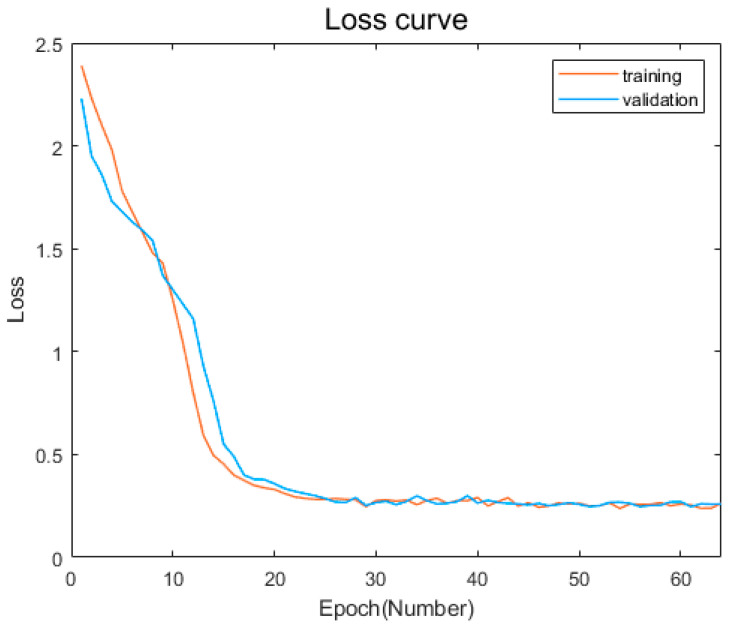
Loss value change curves of training and verification sets.

**Figure 12 sensors-23-05636-f012:**
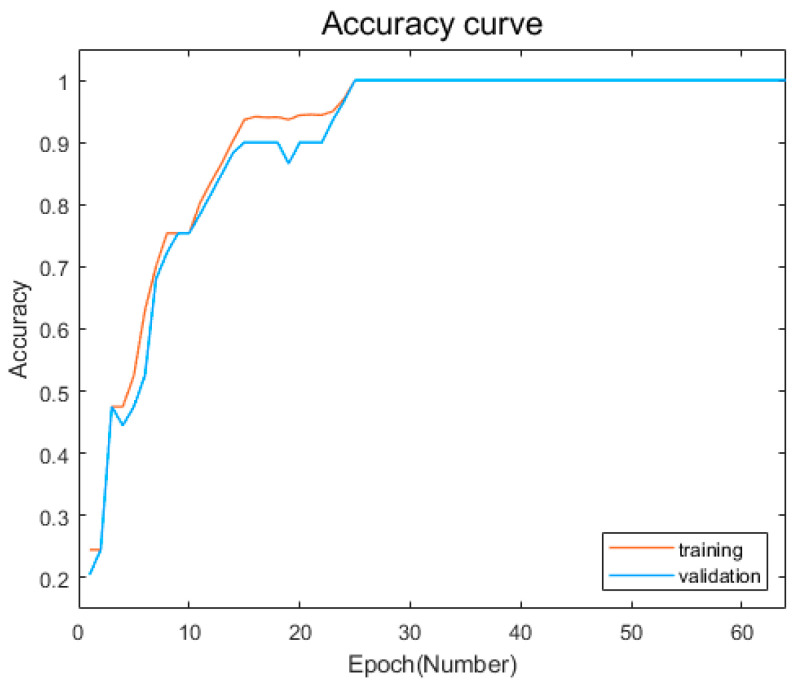
Accuracy change curves of the training and validation sets.

**Figure 13 sensors-23-05636-f013:**
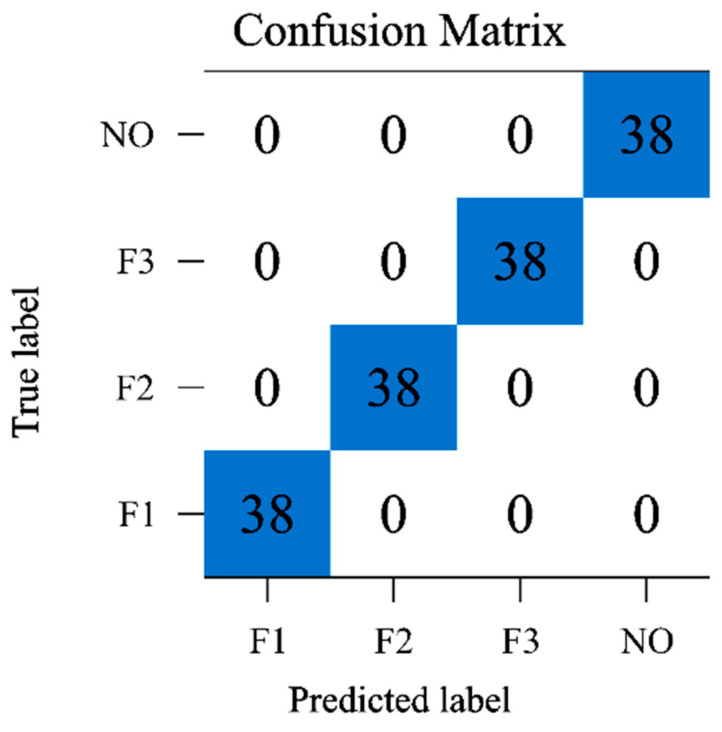
Confusion matrix at t = 2.0 s.

**Table 1 sensors-23-05636-t001:** LRE failure modes.

Components	Classification	Fault Mode	Fault Performance
Turbopump	Centrifugal Pump	(1) Impeller damage (2) Bearing wear or damage (3) Pump cavitation	Pump efficiency decrease
	Turbine	(1) Blade detachment (2) Bearing wear or damage (3) Turbine blade erosion (4) Gas flow obstruction (5) Turbine inlet flow leakage	Turbine efficiency decrease
			Downstream flow rate decrease
Pipeline	Gas pipeline	(1) Pipeline blockage (2) Pipeline leakage	Increased flow resistance
	Liquid pipeline		Downstream flow rate decrease
Thrust chamber	Combustion chamber	Combustion deterioration	Combustion efficiency decrease
	Gas generator	Combustion deterioration	
	Cooling jacket	Cooling jacket blockage	Increased flow resistance
		Cooling jacket leakage	Downstream flow rate decrease
	Nozzle	(1) Nozzle deformation (2) Large nozzle detachment	Nozzle efficiency decrease
Others	Regulating valve	Stuck during switching	Reduced flow area
	Cavitation tube	Cavitation tube blockage	Increased flow resistance
	Sonic nozzle	Sonic nozzle blockage	

**Table 2 sensors-23-05636-t002:** The average accuracy, standard deviation and training time of the four models.

Diagnosis Method	CNN	1DCNN-SVM	CNN-LSTM	1DCNN-A-BiLSTM
Ten times average classification accuracy/%	86.99	93.83	94.76	97.39
Standard deviation/%	2.3236	1.3798	0.6271	0.5832
Time/s (Using CPU)	6.8	9.4	11.3	8.7

## Data Availability

The data is not publicly available as it involves sensitive information.

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
