# Peer review of "Intelligent Fault Diagnosis of Liquid Rocket Engine via Interpretable LSTM with Multisensory Data"

_sensors, 2023, doi:10.3390/s23125636_

Round 1

Reviewer 1 Report

This manuscript proposed an interpretable CNN-LSTM model for fault diagnosis by using the simulated measurement data of the LRE mathematical model. The attention mechanism was adopted for interpretable deep neural network, which would increase the generalization of deep models. This paper has certain novelty and meaningful contribution. But there are still inadequacies that need further improvement. Technical items for which revisions are recommended:

1.      In section 3.2, numbering should start from 1 instead of 4.

2.      The authors should properly describe the Bidirectional LSTM and Attention mechanism using the mathematical formalism.

3.      It would be great if the authors could add a comparison chart or table with the conventional model to illustrate the advantages mentioned in the article.

4.      There are some grammatical errors throughout the text that should be corrected before the paper is accepted.

There are some grammatical errors throughout the text that should be corrected before the paper is accepted.

Reviewer 2 Report

1. Please provide quantitative results in abstract.

2. Please provide relevant citations for Lines 29 to 31.

3. More literature review related to data driven based approaches need to be provided. Less than 10 papers are covered in this manuscript. 

4. Authors need to explain the main differences between their proposed work and those published ones.

5. Please provide full name of Bi-LSTM in Line 18 and EM algorithm in Line 36.

6. Problem statements and research gaps that lead to current research works are not well explained. Please elaborate for better clarity.

7. Both contributions look similar and authors are required to rewrite the contributions for better clarity.

8. Line 114 - "A" in the sentence of "... a maximum flow area of A..." should be presented in italic form to avoid confusion.

9. What are the differences between these two equations of Eqs. (2.4) and (2.5)?

10. P in 130 and n in 140 should be P_turbine and n_turbine instead?

11. One of my major concern is the proposed methodology part because it is poorly presented. Some contents presented in Section 4 (up to Line 305) should be merged with Section 3. 

12. Authors Starts with the overall work flow of proposed method. Then, provide detailed explanation of each step. Appropriate mathematical formulations need to be provided. I notice that many important mathematical equations are not provided by authors while describing their works, particulary for the BiLSTM and Attention module.

13. - Descriptions of datasets and data preprocessing parts are missing. The characteristics of datasets used in current study are not mentioned at all.

14. How the features are extracted by CNN are not mentioned as well.

15. Symbols used in the Figure 2 are not defined in the text. Differences between LSTM_R and LSTM_L should be explained. 

16.  Figure 4 is too small and did not provide any meaningful information. How to differentiate normal and faulty states? What are the meanings of graphs with different colors?

17. Figure 5 is too small and many details cannot be read, therefore it is hard to understand. Need to redraw for better clarity.

18. Performance comparisons are done in vert superficial way. More in-depth performance analysis are needed to justify the strength of proposed work.

19. Authors have mentioned about the interpretable BiLSTM part through Attention mechanism. How is this going to achieve performance gains of proposed work? Please further analyse. 

20. Performance comparisons with state-of-art works should be done to investigate the strenght of proposed work.

21. Source codes of proposed work should be provided to benefits other prospective researchers that are interested in this research area.

Some minor grammatical issues and typo are observed in the manuscript. Authors need to proofread the manuscript again.

Round 2

Reviewer 2 Report

look better now. no major issues raised this round.

english quality is ok.